# REVISITING THE DESIGN CHOICES IN MAX-RETURN SEQUENCE MODELING

## ABSTRACT

Decision Transformer (DT), free from optimal value functions fitting and policy gradient computation, attempts to solve offline reinforcement learning (RL) via supervised sequence modeling. During inference, sequence modeling requires an initial target returns assigned with expert knowledge, which blocks comprehensive evaluation on more diverse datasets. As a result, existing sequence modeling only focuses on limited evaluation on `Gym` datasets and some understanding is severely biased. In this paper, we aim to revisit the design choices, including architecture and context length, in sequence modeling on more diverse datasets. We utilize the max-return sequence modeling that replaces the manual target returns with maximized returns predicted by itself. We systematically investigate the impact of 1) architectural choices and 2) context lengths in max-return sequence modeling on nine datasets with varying data distributions. Abundant experiments and thorough analyses reveal that design choices are highly influenced by the dataset characteristics, which further underscores the significance of more diverse evaluation.

## 1 INTRODUCTION

Classical online reinforcement learning (RL) algorithms such as Q-learning (Watkins & Dayan, 1992) or policy gradient (Sutton et al., 1999) are derived from the Markov Decision Process (MDP) (Sutton et al., 1998) formulation. Sequence modeling (Chen et al., 2021), developed in data-driven offline scenario (Levine et al., 2020; Fu et al., 2020), maximizes the likelihood of actions based on the whole historical trajectories that including state, action and returns. In this way, offline RL is addressed from one paradigm similar to the supervised learning. A particularly enticing prospect is that the successes of supervised sequence modeling in other domains may be replicable within the offline realm, potentially catapulting the rapid advancement and progress of reinforcement learning.

However, the existing evaluation of sequence modeling is insufficient and consequently the corresponding understanding is biased and limited, which has hindered the further development of sequence modeling. The insufficient evaluation stems from the choice of the initial returns target during the sequence modeling inference (Zheng et al., 2022; Chen et al., 2021; Lee et al., 2022). The initial returns target serves as a inference hyperparameter that should be meticulous determined using expert knowledge or extensive experiments. Decision transformer (DT) (Chen et al., 2021) proposes the initial returns targets on D4RL-`Gym` datasets using domain knowledge and online decision transformer (ODT) (Zheng et al., 2022) further optimizes hyperparameters via exhaustive experimental comparison. in contrast, the initial returns targets on other representative datasets (`Antmaze`, `Maze2d`, `Kitchen` and `Adroit`) are under explored. As a result, subsequent research on sequence modeling has been almost exclusively confined to the `Gym`, neglecting other datasets (Zhuang et al., 2024). Even the paper that examines the advantages and disadvantages of DT in comparison to CQL (Kumar et al., 2020) and BC (Pomerleau, 1988) exhibit a similar bias (Bhargava et al., 2023). This status results in an insufficient investigation of the impact of dataset characteristics on sequence modeling, and explorations about historical sequence length and sequence model architecture are consistently biased.

In this paper, we aim to systematically investigate the impact of 1) dataset characteristics 2) architectural choices and 3) the length of historical sequences on the performance of sequence modeling. To overcome the limitation of the human designed initial returns targets, we adopt the max-return sequence modeling introduced by Reinforced Transformer (Rein*for*mer) (Zhuang et al., 2024). The

fundamental premise of max-return sequence modeling is to bring the concept of return maximization back to the supervised paradigm of sequence modeling. In terms of implementation, max-return sequence modeling predicts a maximized return at each timestep to guide the generation of actions, free from specifying an initial returns target. We have conducted exhaustive experiments and analytical studies, leading to the following conclusions and findings:

- Overall, the dataset characteristics of the have greater an impact on sequence modeling than the model architecture and context length. Discussing the impact of other factors without considering dataset characteristics is quite one-sided. In trajectory stitching problems, sequence modeling is inherently at a disadvantage compared to RL algorithms (Brandfonbrener et al., 2022). Sequence modeling is more adept at long-term tasks and tasks that include part of expert data.

- In terms of architecture, the Rein*for*mer tends to consider global information, while Reinconver (Reinforced Convformer) and Reimba (Reinforced Mamba) focus more on local information.

- The impact of context length on performance is relatively minor. A shorter context length is more advantageous for trajectory stitching. Moreover, it is surprising to find that models trained on long sequences perform exceptionally well during inference with short sequences, significantly enhancing their trajectory stitching capabilities.

## 2 PRELIMINARY

### 2.1 OFFLINE REINFORCEMENT LEARNING

Offline RL (Levine et al., 2020) forbids the interaction with the environment and only a fixed offline dataset full of trajectories $\mathcal{D} = \{(s_0, a_0, r_0, s_1, a_1, r_1, \cdots, s_t, a_t, r_t \cdots)\}$ is provided . Here $s_t$ is the current state at timestep $t$, $a_t$ is the action and $r_t \dot{=} r(s_t, a_t)$ is the reward of current state and action. The objective of offline RL is to learn a policy $\pi(a_t|s_t)$ that maximizes the expected returns $\mathbb{E}_\pi \left[ \sum_{t=0}^T r(s_t, a_t) \right]$. Compared to the traditional online RL (Sutton et al., 1998), this setting is more challenging since the agent is unable to explore the environment and collect extra feedback.

### 2.2 SEQUENCE MODELING

Sequence modeling (Chen et al., 2021) breaks the traditional Markov property and the prediction of the current action $a_t$ is based on the entire historical trajectories $\tau_{t-K}$:

$$\tau_{t-K} = (R_{t-K+1}, s_{t-K+1}, a_{t-K+1}, \cdots, R_{t+1}, s_{t+1}, a_{t+1}), \tag{1}$$

where $R_t \dot{=} \sum_{t'=t}^T r_t$ is called returns-to-go (or simply returns) that represents the sum of future rewards from current timestep $t$. $\tau_{t-K}$ contains the previous $K$ timesteps trajectory and $K$ is called context length. Sequence modeling directly maximizes the likelihood of actions conditioned on not only the current state $s_t$ and returns-to-go $R_t$, but also the historical trajectories $\tau_{t-K}$:

$$\mathcal{L}_{\text{DT}} = -\mathbb{E}_t \left[ \log \pi(a_t|\tau_{t-K}, s_t, R_t) \right]. \tag{2}$$

This training loss (2) indicates offline RL is solved from the perspective of supervised learning, rather than traditional RL paradigm. Besides, the implementation of $\pi$ is based on sequence models such as transformer (Vaswani et al., 2017). For the **Inference**, the initial target returns $\hat{R}_0$ should be determined first. Given $\hat{R}_0$ and the initial environment state $s_0$, the next action will be generated by the model $\pi\left(a_1|\hat{R}_0, s_0\right)$. Once the action $a_1$ is executed by the environment, the next state $s_1$ and reward $r_1$ are returned. Then the next returns-to-go should minus the returned reward $\hat{R}_1 = \hat{R}_0 - r_1$. This process is repeated until the episode terminates.

## 3 BACKGROUND AND EXPERIMENTAL SETUP

In this section, we first provide an overview of the current insufficient evaluation of sequence modeling and analyze the reason behind it. We then introduce the max-return sequence modeling and discuss why the concept of max-return has the potential to ameliorate this situation. Finally, we detail our specific experimental setup.

## 3.1 INSUFFICIENT EVALUATION

We summarize the datasets that have been evaluated by representative sequence modeling algorithms in the Table 1. Obviously, all the methods consider the D4RL-`Gym` datasets, while other datasets with various characteristics are ignored[1]. For example, `Antmaze-medium` datasets require the algorithm to wisely stitch the sub-optimal trajectories into successful ones to achieve the final goal. This phenomenon is called trajectory stitching ability and usually, RL is believed to possess this ability inherently while sequence modeling not. Algorithms aimed for trajectory stitching should be evaluated on these datasets, yet these datasets are overlooked (Wu et al., 2023).

Table 1: Insufficient Evaluation of Sequence Modeling.

|  | Gym | Antmaze-u | Antmaze-m | Maze2d | Kitchen | Adroit |
|---|---|---|---|---|---|---|
| DT (Chen et al., 2021) | ✓ | ○ | ○ | ○ | ○ | ○ |
| ODT (Zheng et al., 2022) | ✓ | ✓ | ○ | ○ | ○ | ○ |
| EDT (Wu et al., 2023) | ✓ | ○ | ○ | ○ | ○ | ○ |
| DC (Kim et al., 2023) | ✓ | ✓ | ○ | ○ | ○ | ○ |
| DS4 (David et al., 2022) | ✓ | ✓ | ○ | ○ | ○ | ○ |
| DM (Lv et al., 2024) | ✓ | ✓ | ○ | ○ | ○ | ○ |
| DMamba (Ota, 2024) | ✓ | ○ | ○ | ○ | ○ | ○ |
| DM-H (Huang et al., 2024) | ✓ | ○ | ○ | ○ | ○ | ○ |
| MambaDM (Cao et al., 2024) | ✓ | ○ | ○ | ○ | ○ | ○ |
| Rein*for*mer (Zhuang et al., 2024) | ✓ | ✓ | ✓ | ✓ | ✓ | ○ |

The underlying reason behind insufficient evaluation is the relative difficulty in selecting the hyperparameter initial target returns $\hat{R}_0$ during inference. An inappropriate choice of $\hat{R}_0$ may hinder the model from realizing its full potential. The selection of this hyperparameter on the `Gym` datasets is first introduced by DT (Chen et al., 2021) and later optimized by ODT (Zheng et al., 2022) through extensive experiments. But the choice of $\hat{R}_0$ for other datasets receives little attention. Rein*for*mer (Zhuang et al., 2024) proposes to replace this expert-designed returns target with predicted maximized return, overcoming the challenge of $\hat{R}_0$ selection.

## 3.2 MAX-RETURN SEQUENCE MODELING

Since supervised sequence modeling does not explicitly consider return maximization, the core objective of RL, the concept of max-return sequence modeling is introduced. The key lies in utilizing the maximized return to guide the generation of next actions during inference.

Concretely, Reinforced Transformer (**Rein*for*mer**) adopt the following historical trajectories $\tau_{t-K}$:

$$\tau_{t-K} = (s_{t-K+1}, R_{t-K+1}, a_{t-K+1}, \cdots, s_{t+1}, R_{t+1}, a_{t+1}),\tag{3}$$

where the state $s_t$ is placed before the returns-to-go $\hat{R}_t$, different from the original formulation 3. The most significant advantage is that the reward can be predicted through the state without the need for prior specification. During the training phase, in addition to the loss function that maximizes the action probability, **Rein*for*mer** also introduces a return loss based on the expectile regression:

$$\mathcal{L}_{\text{Rein}for\text{mer}} = \mathbb{E}_t \left[ -\log \pi\left(a_t | \tau_{t-K}, s_t, R_t\right) + |\alpha - \mathbb{1}\left(\Delta R_t < 0\right)| \Delta R_t^2 \right],\tag{4}$$

where $\Delta R_t = R_t - \pi\left(\hat{R}_t | \tau_{t-K}, s_t\right)$ is the difference between the oracle return $R_t$ and its prediction $\hat{R}_t$. Here $\alpha \in (0, 1)$ is the hyperparameter of expectile regression. When $\alpha = 0.5$, expectile regression degenerates into standard MSE loss. But when $\alpha > 0.5$, this asymmetric loss will give more weights to the $R_t$ larger than $\hat{R}_t$. Furthermore, it can be proved that this additional return loss function can make the model predict the maximum returns-to-go when $\alpha \to 1$, which is similar to the maximizing returns objective in RL. For the **inference**, the maximized initial target returns $\pi\left(\hat{R}_0 | s_0\right)$ is predicted given the initial environment state $s_0$ rather than manually designated. Since the $\hat{R}_0$ is maximized, the next action $\pi\left(a_1 | \hat{R}_0, s_0\right)$ will approach to the optimal one. Then the next state $s_1$ is returned and this process is repeated until the episode terminates.

---

[1]Strictly, Q-value Regularized Transformer (QT) (Hu et al., 2024) does not belong to sequence modeling since it requires Q value as part of gradient

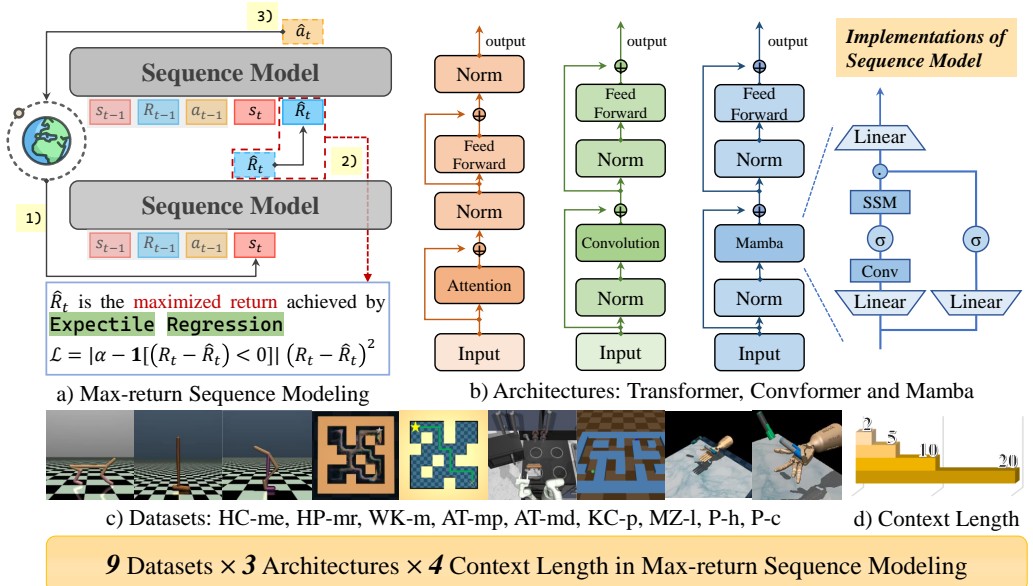

Figure 1: The figure shows the overview of this paper. a) Max-return Sequence Modeling: During inference, the first step involves predicting the maximized return using expectile regression, aiming to select trajectories in the dataset with the maximum return to go in the current state sequence. In the second step, the predicted returns is reintroduced back into the same transformer. The key difference here compared to the first step is that an additional token is included in the transformer's input. It is at this stage that we obtain the desired action. Totally, b) **3** architectures, c) 9 datasets and d) **4** context lengths are considered in our experiments.

### 3.3 EXPERIMENTAL SETTING

Now, we are ready to systematically investigate the impact of 1) dataset characteristics 2) architectural choices and 3) context length on the performance of max-return sequence modeling. The Overview of the experimental setting is summarized in Figure 1.

**Dataset Characteristics** We selected nine representative datasets from the widely-used offline benchmark D4RL to evaluate the sequence modeling, which are detailed as follows:

- `Halfcheetah-medium-expert`, `Hopper-medium-replay` and `Walker2d-medium`: The abbreviations are respectively `HC-me`, `HP-mr` and `WK-m`. For `Gym` tasks, we select only one dataset from each environment, which encompasses three distinct data distributions. The "medium-replay" dataset consists of samples in the replay buffer observed during online training until the policy reaches the "medium" level, approximately 1/3 the performance of the "expert".

- `Antmaze-medium-play` and `Antmaze-medium-diverse`: The abbreviations are respectively `AT-mp` and `AT-md`. `Antmaze` datasets have a sparse reward to show if the ant reach the goal in the maze. The `medium` dataset requires the algorithm to navigate to the target point by stitching the suboptimal trajectories into the successful trajectories. These datasets require the trajectory stitching ability, which is particularly challenging for sequence modeling.

- `Kitchen-partial`: The abbreviation is `KC-p`. The desired goals are to complete 4 subtasks: open the microwave, move the kettle, flip the light switch, and slide open the cabinet door. The "partial" dataset includes subtrajectories where the 4 target subtasks are completed in sequence.

- `maze2d-large`: The abbreviation is `MZ-l`. The dataset is collected by a PD controller that memorizes the reached waypoints during data collection, so the Markov property does not hold.

- `Pen-human` and `Pen-cloned`: The abbreviations are respectively `P-h` and `P-c`. This environment controls a 24-DoF simulated Shadow Hand robot to twirl a pen. `Human` dataset contains 25 trajectories of expert demonstration. `Cloned` dataset uses a 50-50 split between demonstration data and trajectories sampled from a behavior cloned policy trained on the demonstrations.

In summary, these 9 datasets each have their own distinctive features. In addition to the three commonly used `Gym` datasets, our selection also encompasses `Antmaze` datasets that emphasize trajectory stitching, `Kitchen` dataset that includes partial expert demonstration segments, `maze` dataset highlighting non-Markovian properties, and `Pen` dataset that incorporates expert demonstrations.

**Architectural Choices**  To accommodate the max-return sequence modeling, we employ the following inputs and outputs:

$$\textbf{Input:}\ \ \langle \tau_{t-K}, s_t, g_t \rangle \xrightarrow{\pi} \textbf{Output:}\ \ \langle \hat{g}_t, \hat{a}_t \rangle. \tag{5}$$

The implementation of policy $\pi$ is based on the sequence model and the predictions $\hat{g}_t, \hat{a}_t$ are achieved through an autoregressive approach. Moving forward, we primarily consider three architectural variants: the Transformer (Vaswani et al., 2017), One-dimensional convolutional layers (Conv) (Yu et al., 2022), and the linear Recurrent Neural Network Mamba (Gu & Dao, 2023).

- Rein*for*mer is based on the Transformer architecture, built upon the self-attention mechanism, equipped with multiple attention heads and stacked encoder-decoder structures, can adeptly captures long-range dependencies. The Decoder module within the Transformer has found wide application in NLP and Offline Reinforcement Learning tasks, as demonstrated by models like Decision Transformer. The equation presented exemplifies the attention mechanism used in the Transformer framework:

$$Attention(Q, K, V) = softmax\left(\frac{QK^T}{\sqrt{d_k}}\right)V \tag{6}$$

- Reinconver is based on 1D CNN. In the field of sequence modeling, 1D convolutions play a role in extracting local patterns and features from sequences, aiding in learning positional invariance. It is worth mentioning that, positional information is inherently included during the convolution process due to the local receptive field property, so we did not add positional embedding to Reinconver.

- Reimba is based on the linear RNN Mamba. Inspired by continuous-time systems, Mamba models sequences or one-dimensional functions through a recurrent mapping process. Like S4, Mamba uses a hidden state representation, where the hidden state evolves through time as the system processes inputs. These equations describe the time evolution of the hidden state, with:

$$h'(t) = Ah(t) + Bx(t), \quad y(t) = Ch(t), \tag{7}$$

where $A \in \mathbb{R}^{N \times N}$ is the evolution matrix, $B \in \mathbb{R}^{N \times 1}$ and $C \in \mathbb{R}^{1 \times N}$ are projection matrices that govern how inputs and hidden states are transformed into outputs. In the discrete case, Mamba uses techniques similar to S4, where continuous parameters $A$ and $B$ are discretized, enabling the model to handle sequences. This leads to a discrete-time variant of the ODEs:

$$h_t = \bar{A}h_{t-1} + \bar{B}x_t, \quad y_t = Ch_t, \tag{8}$$

where $\bar{A} = \exp(\Delta)A$ and $\bar{B} = (\Delta A)^{-1}(\exp(\Delta A) - I)(\Delta B)$, with $\Delta$ representing a timescale parameter. Mamba introduces a selective scan mechanism, allowing it to dynamically evolve hidden states based on input data, which ensures Mamba efficiently captures long-range dependencies while maintaining computational efficiency for long sequences. Mamba is currently a hot contender in the fields of CV and NLP. At the same time, since Mamba is essentially a type of RNN-like structure capable of extracting positional information, we did not include positional embedding to Reimba.

**Context Length**  Traditional offline RL algorithms, derived from Markov Decision Processes (MDPs), predict actions solely based on the current state $s_t$. In contrast, sequence modeling also takes into account historical trajectories of length $K$, thereby breaking the Markov property. Thus, the context length is a factor worthy of exploration in sequence modeling.

We consider 4 context length $K = 2, 5, 10, 20$, with the maximum value of 20 being the default context length for DT, and the minimum of 2 corresponding to the shortest sequence length. The intermediate values are determined by exponential interpolation. Typically, the context length during inference should be consistent with the training training. However, there are exceptions, such as ODT (Zheng et al., 2022), which manually adjusts the sequence length during inference, and EDT (Wu et al., 2023), which dynamically adjusts K based on whether the current trajectory is optimal.

## 4 RELATED WORK

Offline Reinforcement Learning (Levine et al., 2020) breaks free from the traditional paradigm of online interaction (Sutton et al., 1998) and learns policy from fixed offline dataset collected by arbitrary or even unknown process (Lange et al., 2012; Fu et al., 2020). Most offline RL algorithms are developed based on classical online algorithms, such as CQL (Kumar et al., 2020) based on SAC (Haarnoja et al., 2018), TD3+BC (Fujimoto & Gu, 2021) based on TD3 (Fujimoto et al., 2018) and BPPO (Zhuang et al., 2023) based on PPO (Schulman et al., 2017). In contrast, Decision Transformer (DT) (Chen et al., 2021) directly maximizes the action likelihood, solving offline RL from supervised sequence modeling paradigm. Following upside-down RL (Srivastava et al., 2019; Schmidhuber, 2019), DT considers returns when predicting the action. Some works equip DT with classical RL components including dynamics programming (Yamagata et al., 2023), critic guidance (Wang et al., 2024; Hu et al., 2024), return maximization (Zhuang et al., 2024), online finetuning (Zheng et al., 2022) and trajectory stitching (Wu et al., 2023). On the other hand, DT is investigated from supervised learning perspective such as unsupervised pretraining (Xie et al., 2023; Carroll et al., 2022) and scaling ability (Lee et al., 2022; Shridhar et al., 2023). As for model architecture, LSTM (Siebenborn et al., 2022), one-dimension convolution network (Kim et al., 2023; Yan et al., 2024) and linear RNN (David et al., 2022; Cao et al., 2024; Ota, 2024; Lv et al., 2024; Huang et al., 2024) are adopted to replace the transformer (Vaswani et al., 2017) in DT. However, the evaluation of these sequence models is based on limited datasets, including only the Gym and antmaze-umaze datasets (Fu et al., 2020), which biases the drawn conclusions. Rein*for*mer (Zhuang et al., 2024) proposes using maximized returns during inference to replace the manually designed initial target returns in DT, significantly expanding the range of evaluated datasets. This paper aims to leverage the max-return sequence modeling proposed by the Rein*for*mer (Zhuang et al., 2024) to conduct a systematic evaluation of sequence models and reveal the future direction.

## 5 RESULTS AND DISCUSSION

In this section, we present the performance of max-return sequence modeling across 9 datasets, 3 architectures, and 4 different context lengths, followed by an in-depth analysis. Specifically, we focus on three key questions: 1) What types of data distributions are suitable for sequence modeling? 2) What characteristics do different architectures exhibit? 3) How does the training and inference length of historical trajectories affect performance?

### 5.1 MAIN RESULTS

Table 2: The normalized score of max-return sequence modeling on 9 datasets (`HC-me`, `HP-mr`, `WK-m`,`AT-mp`, `AT-md`, `KC-p`, `MZ-l`, `P-h`, `P-c`) with 3 different architectures and 4 context-lengths ($K$). We report the mean of normalized score for five seeds. For each seed, the normalized score is calculated by the mean of 10 evaluation trajectories for `Gym` and `Adroit` while 100 for `Antmaze`, `Maze2d` and `Kitchen`. We also compare our result with IQL, highlighting scores below IQL in gray. The best result is red and the **bold** result means the best result among one sequence model with different $K$. The last row represents how many results outperforms IQL.

| model | $K$ | HC-me | HP-mr | WK-m | AT-mp | AT-md | KC-p | MZ-l | P-h | P-c |
|---|---|---|---|---|---|---|---|---|---|---|
| Rein*for*mer | 2 | 91.23 | **70.92** | 79.84 | **5.80** | 2.00 | 68.05 | NaN | 62.77 | 64.49 |
| Rein*for*mer | 5 | 90.99 | 68.80 | **79.91** | 4.20 | 3.40 | 73.00 | 64.95 | **75.15** | **86.55** |
| Rein*for*mer | 10 | 91.87 | 53.02 | 79.82 | 3.80 | **5.60** | **74.05** | 62.00 | 68.25 | 75.17 |
| Rein*for*mer | 20 | **92.81** | 40.84 | 72.25 | 1.60 | 4.20 | 66.20 | **64.99** | 71.94 | 74.79 |
| Reinconver | 2 | 91.83 | 84.24 | 72.28 | 6.20 | **5.40** | 65.20 | 32.69 | 73.64 | 68.52 |
| Reinconver | 5 | 92.26 | **84.44** | 74.09 | **7.80** | 4.20 | 34.55 | 22.45 | **82.23** | 71.68 |
| Reinconver | 10 | **92.90** | 54.02 | **75.88** | 4.40 | 5.20 | **65.85** | 34.74 | 76.27 | 62.58 |
| Reinconver | 20 | 92.78 | 49.22 | 75.38 | 2.00 | 2.60 | 65.25 | 32.39 | 75.29 | **83.38** |
| Reimba | 2 | 91.79 | **81.95** | 77.81 | 5.20 | 2.60 | 40.75 | **59.00** | 84.89 | 59.60 |
| Reimba | 5 | 92.91 | 74.24 | **80.03** | 12.40 | 5.00 | **45.10** | 41.04 | 82.91 | **71.28** |
| Reimba | 10 | **93.05** | 55.99 | 75.59 | 13.80 | 5.00 | 29.70 | 43.59 | **97.31** | 71.02 |
| Reimba | 20 | 92.42 | 49.47 | 73.35 | **15.60** | **9.00** | 29.05 | 43.14 | 91.61 | 70.57 |
| IQL | 1 | 86.70 | **94.70** | 78.30 | **78.50** | **83.50** | 46.30 | 61.70 | 71.50 | 37.30 |
| | | (12/12) | (0/12) | (4/12) | (0/12) | (0/12) | (6/12) | (3/12) | (10/12) | (12/12) |

Table 2 presents the performance of max-return sequence modeling with different parameters on diverse data distribution. We also compare the performance of sequence modeling algorithms with the classic offline reinforcement learning algorithm, IQL, highlighting scores below IQL in gray. Although IQL no longer represents the current state-of-the-art (SOTA) offline algorithm, it still significantly outperforms sequence modeling on some datasets.

First of all, with datasets that contains high-quality data (such as `HC-me`, `WK-m`) or even expert demonstration (such as `P-h`, `P-c`), max-return sequence modeling often excels IQL. Although max-return sequence modeling introduces the concept of return maximization, it also resembles to supervised learning that prefers high-quality data. Second, when faced with low-quality datasets (such as `HP-mr`,`AT-mp`, `AT-md`), RL maintains an overwhelming advantage. In the subsequent analysis and discussions on architecture and context length, we also focus on these three datasets.

## 5.2 ARCHITECTURE

In this section, we explore the impact of model architecture on the performance of sequence modeling. Prior research has indicated that the sequence modeling with Convformer (Kim et al., 2023) and Mamba architecture (Ota, 2024; Cao et al., 2024; Lv et al., 2024; Huang et al., 2024) outperform classical transformer. However, these conclusions were drawn without considering the data distribution and its characteristics. Therefore, we will re-examine these findings across 9 datasets.

### 5.2.1 ATTENTION ON HISTORICAL TOKENS

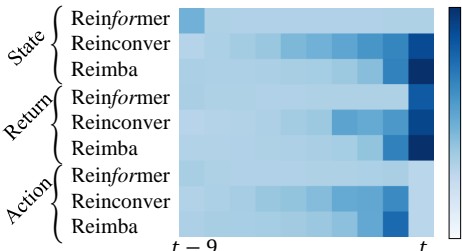

Figure 2: This heatmap illustrates the impact of token zero masking on the final output.

We analyze which part of the historical trajectory different model constructions specifically focus on. We selected the model trained with $K = 10$ on the Antmaze environment. Let $t$ represent the time step of the current token, and $t - 9$ represents the token furthest from the current time step. By masking a token at a certain position with 0, we calculate the difference between the masked output and the original output. This difference can, to some extent, reflect the importance of the masked token to the current output value. Then, based on this difference, we can determine whether the model pays more attention to global or local information. We have plotted the heatmaps of the differences in state, return, and action for the three models in the right figure.

The heatmap reveals that Reinconver and Reimba exhibit a significant increase in values at the current timestep, indicating a greater focus on local information. In contrast, the Rein*for*mer does not show a marked rise in differences, suggesting that the impact of masking any token at the current timestep is relatively uniform. Thus, the Reinformer pays more attention to global information.

### 5.2.2 ARCHITECTURE COMPARISON

In Figure 3, we illustrate the probability of the architecture on the left outperforming the right one. The closer the box is to the right side, the better the performance of the model on the left, and vice versa. A position in the center indicates that the two architectures have comparable performance. Considering the nine datasets collectively, no single model demonstrates an absolute advantage. In other words, the superiority of a model cannot be discussed independently of the characteristics of the dataset.

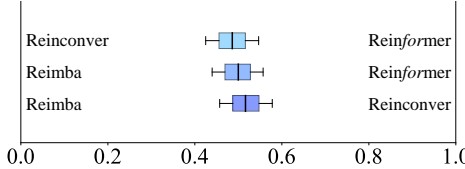

Figure 3: The improvement probability of the architectures across all the 9 datasets.

On the maze-large dataset, the Reinformer demonstrates a significant advantage. This is because the maze-large dataset inherently exhibits non-Markovian properties, where decisions based on the current state are correlated with historical waypoints. The Reinformer's focus on global historical trajectory information is particularly adept at considering and utilizing waypoint-related information effectively. In contrast, on the Antmaze-medium-play dataset, which emphasizes trajectory stitching, models like Reincover and Reimba that focus on local information perform better. This is

attributed to the fact that extensive historical sequence information leads to more conservative model outputs, reducing the likelihood of generating new decisions that deviate from historical trajectories.

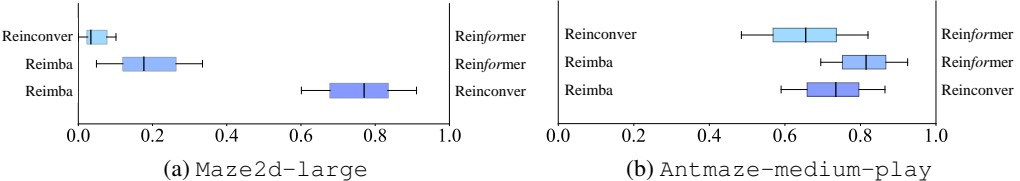

(a) `Maze2d-large`       (b) `Antmaze-medium-play`

Figure 4: The probability of the model on the left superior to the model on the right across (a) `Maze2d-large` and (b) `Antmaze-medium-play`.

### 5.2.3 INFLUENCE OF POSITIONAL EMBEDDING

As previously mentioned, we do not use positional embedding in Reinconver and Reimba. We believe the positional embedding is harmful to trajectory stitching. Positional embedding are directly added to embedded state, returns and action tokens. As a result, the same input sequences become different at different timesteps, which is harmful to stitching under similar state sequences. This is supported by "w/o" results in Table 3 especially on short Context Length.

Table 3: The normalized scores of Reinconver and Reimba without and with positional embedding. Default Reimba and Reinconver did not include positional embedding.

| model | $K$ | HP-mr | | | AT-mp | | |
|---|---|---|---|---|---|---|---|
| | | w/o | w/ | $\Delta$ | w/o | w/ | $\Delta$ |
| Reinconver | 2 | 84.24 | 67.55 | -19.81% | 6.20 | 2.67 | -56.94% |
| Reinconver | 5 | 84.44 | 72.74 | -13.86% | 7.80 | 7.00 | -10.26% |
| Reinconver | 10 | 54.02 | 75.52 | +39.80% | 4.40 | 2.33 | -47.05% |
| Reinconver | 20 | 49.22 | 68.87 | +39.92% | 2.00 | 1.00 | -50.00% |
| Reimba | 2 | 81.95 | 76.87 | -6.20% | 5.20 | 8.33 | +60.19% |
| Reimba | 5 | 74.24 | 77.23 | +4.03% | 12.40 | 11.00 | -11.29% |
| Reimba | 10 | 55.99 | 60.94 | +8.84% | 13.80 | 10.00 | -27.54% |
| Reimba | 20 | 49.47 | 70.03 | +41.56% | 15.60 | 11.00 | -29.49% |

Positional embedding facilitates effective information extraction from long sequences. On `Hp-mr` dataset, the advantage of long sequences with positional embedding in information extraction outweighs their disadvantage in trajectory stitching, causing performance improvement with large $K$. But on `AT-mp` that heavily emphasizes stitching, the advantage in information extraction does not surpass the disadvantage in trajectory stitching, even in the scenario of large $K$.

### 5.3 CONTEXT LENGTH

In this subsection, we investigate the impact of the historical sequence length, also known as context length, on performance. Sequence modeling and Markov Decision Processes (MDPs) have distinct perspectives on the historical trajectory when predicting actions, making context length a crucial factor in sequence modeling.

### 5.3.1 CONTEXT LENGTH COMPARISON

In Table 4, we employ the least squares method to calculate the fitted line of performance with respect to $K$. And the slope of the fitted line is extracted to describe the relationship between the performance and context length $K$ across various datasets. Overall, performance fluctuates

Table 4: The slope of the fitted line from the least squares method to calculate the fitted line of performance with respect to $K$. This can determine the impact of $K$ on the Normalized Score.

| | HC-me | HP-mr | WK-m | AT-mp | AT-md | KC-p | MZ-l | P-h | P-c |
|---|---|---|---|---|---|---|---|---|---|
| Reinformer | +0.56 | -10.60 | -2.39 | -1.30 | +0.88 | -0.45 | +0.02 | +2.06 | +1.95 |
| Reinconver | +0.35 | -13.55 | +1.11 | -1.60 | -0.74 | +3.15 | +1.14 | -0.10 | +3.55 |
| Reimba | +0.20 | -11.57 | -1.08 | +3.26 | +1.92 | -5.05 | -4.50 | +3.46 | +3.27 |

minimally with changes in $K$, except on `HP-mr` dataset. The performance of sequence modeling notably declines with an increase in context length $K$.

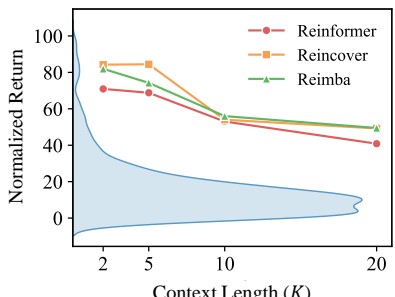

Figure 5: The data distribution (blue shade) and normalized evaluation score on `Hopper-medium-replay`.

We plot the performance curves and the return distribution on `HP-mr` in Figure. The quality of `HP-mr` is widely distributed, ranging from random to expert, with a peak less than 20 normalized score. The distribution of `HP-mr` is akin to an online replay buffer, which places higher demands on learning from suboptimal trajectories. Correspondingly, a smaller context length aligns more closely with the Markov Decision Process (MDP) framework, and thus performs better.

Upon considering all datasets, it becomes evident that no context length is universally applicable across 9 datasets (Figure 6a). For high-quality datasets, such as `HC-me`, a longer context length facilitates better training convergence and ultimately leads to improved score in Figure 6c. For tasks that require trajectory stitching, shorter trajectories are preferred 6b. Taking into account longer historical trajectories increases the influence of past actions on subsequent behaviors, which may hinder the adoption of trajectories that deviate from historical ones. This is detrimental to the stitching process. In other words, longer historical trajectories can also be seen as the conservatism.

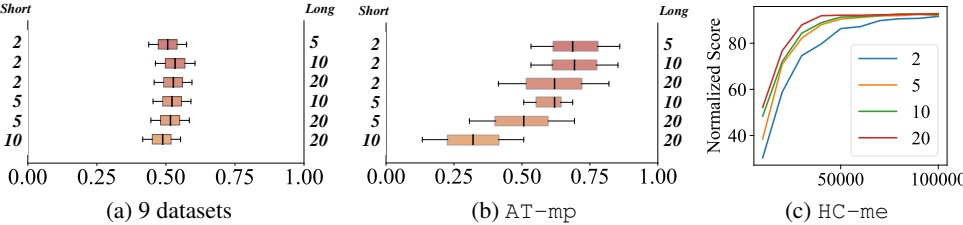

(a) 9 datasets (b) `AT-mp` (c) `HC-me`

Figure 6: (a) represents the probability of the left $K$ superior to the right one across all datasets. (b) represents the probability on `AT-mp` (c) represents evaluation scores with different $K$ on `HC-me`.

### 5.3.2 LONG TRAINING CONTEXT LENGTH WHILE SHORT INFERENCE CONTEXT LENGTH

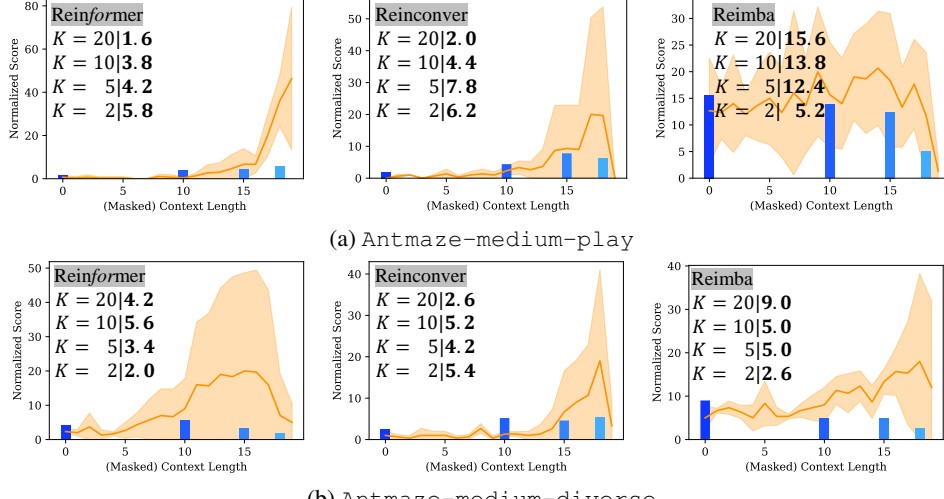

Figure 7: This figure displays the performance of masking the first $(20 - K_1)$ tokens in a sequence model with $K = 20$. We show the averages and corresponding standard deviations of three seeds evaluated in the environment 100 times (represented by the solid yellow line and its shaded area). Additionally, we compare this with models trained and evaluated normally with a length of $20, 10, 5, 2$ (blue bar values). The horizontal axis increases from left to right as the number of masked tokens increases and the remaining context length $K_1$ decreases.

All previous models have considered the historical trajectory length to be the same during inference as in the training phase. ODT (Zheng et al., 2022) discovers that, in some cases, a shorter sequence length during inference can help improve performance. EDT (Wu et al., 2023) also proposes the concept of dynamically adjusting the sequence length during inference based on the quality of the historical trajectory. Therefore, we explore the performance of masking some historical tokens with the model trained by context length $K = 20$.

During the inference phase, we consider historical trajectories of length $K_1 < 20$, padding the empty tokens to with zeros to accommodate the model's input requirements. This can also be interpreted as masking a trajectory segment of length $20 - K_1$ with zeros. In Figure 7, as the horizontal axis increases, the input trajectory length $K_1$ becomes shorter, with the length masked by zeros increasing correspondingly. Concurrently, the model's performance significantly improves, surpassing all the performance with different $K$.

To investigate the reason of significant performance improvement after mask, we conduct the experiments in Figure 8 using the Rein*for*mer with $K = 20$. $x + Att \cdot V$ is generated from normal input while $\widetilde{x} + \widetilde{Att} \cdot \widetilde{V}$ is obtained by masking the first 18 tokens. Then their attention matrices and value vectors are exchanged to obtain $\widetilde{x} + \widetilde{Att} \cdot V$ and $\widetilde{x} + Att \cdot \widetilde{V}$. According to Table 5, replacing $\widetilde{V}$ with $V$ is even worse than normal setting while attention matrices exchange not affect the final performance. This results suggest the Attention matrix $\widetilde{Att}$ obtained after masking did not significantly enhance performance, while $\widetilde{V}$ obtained after masking is the key factor.

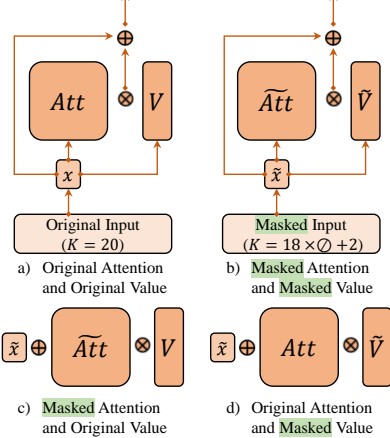

Figure 8: Different Attention matrices and different V inputs are used in the first layer of the transformer to obtain the final action.

Table 5: Result of the experiment showed in figure 8. We report the mean and std of normalized score for three seeds. For each seed, the stats is calculated by 100 evaluation trajectories.

| input | $x + Att \cdot V$ | $\widetilde{x} + \widetilde{Att} \cdot \widetilde{V}$ | $\widetilde{x} + \widetilde{Att} \cdot V$ | $\widetilde{x} + Att \cdot \widetilde{V}$ |
|---|---|---|---|---|
| Normalized Score | 1.60±0.55 | 36.00±11.79 | 0.00±0.00 | 33.00±13.89 |

## 6 CONCLUSION, DISCUSSION AND FUTURE WORK

In this paper, we systematically revisit the impact of 3 architectural choices and 4 context lengths on 9 diverse datasets in max-return sequence modeling. Through extensive experiments, we find:

- Architectures: On more diverse datasets, the Transformer architecture still remains a competitive model. Rein*for*mer exhibits better stability in the presence of input perturbations, focusing more on global sequence information. In contrast, Reinconver and Reimba more focus on local information. The inclusion of positional embedding may not be advantageous for trajectory stitching but aids in information extraction in long sequences.

- Context lengths: In scenarios with high data quality, sequence modeling often outperforms classical offline RL derived from MDPs. However, on datasets that heavily emphasize stitching, classical offline RL surpasses sequence modeling. With exceptionally high data quality, longer sequence models converge faster. Our astonishing discovery is that masking out a portion of historical trajectory information during inference may enhance trajectory stitching.

In summary, we recommend using sequence modeling when data quality is high, resorting to classical Offline RL or a combination of sequence modeling and classical Offline RL when stitching data is crucial. In the future, we will explore how to better integrate classical RL with sequence modeling to harness both of their strengths.

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

## A  HYPERPARAMETERS AND SUPPLEMENTED EXPERIMENTS

Hyperparameters used during model training are as follows:

| env name | model | $K$ | tau | train step | learning rate | normalized score |
|---|---|---|---|---|---|---|
| HC-me | Rein*for*mer | 2 | 0.99 | 10w | 0.0001 | 91.23 |
| | Rein*for*mer | 5 | 0.99 | 10w | 0.0001 | 90.99 |
| | Rein*for*mer | 10 | 0.99 | 10w | 0.0001 | 91.87 |
| | Rein*for*mer | 20 | 0.99 | 10w | 0.0001 | 92.81 |
| | Reinconver | 2 | 0.99 | 10w | 0.0001 | 91.83 |
| | Reinconver | 5 | 0.99 | 10w | 0.0001 | 92.26 |
| | Reinconver | 10 | 0.99 | 10w | 0.0001 | 92.9 |
| | Reinconver | 20 | 0.99 | 10w | 0.0001 | 92.78 |
| | Reimba | 2 | 0.99 | 10w | 0.0001 | 91.79 |
| | Reimba | 5 | 0.99 | 10w | 0.0001 | 92.91 |
| | Reimba | 10 | 0.99 | 8w | 0.0001 | 93.05 |
| | Reimba | 20 | 0.99 | 4w | 0.0001 | 92.42 |
| HP-mr | Rein*for*mer | 2 | 0.999 | 9w | 0.0004 | 70.92 |
| | Rein*for*mer | 5 | 0.999 | 9w | 0.0004 | 68.80 |
| | Rein*for*mer | 10 | 0.999 | 9w | 0.0004 | 53.02 |
| | Rein*for*mer | 20 | 0.999 | 9w | 0.0004 | 40.84 |
| | Reinconver | 2 | 0.999 | 8w | 0.0004 | 84.24 |
| | Reinconver | 5 | 0.999 | 8w | 0.0004 | 84.44 |
| | Reinconver | 10 | 0.999 | 8w | 0.0004 | 54.02 |
| | Reinconver | 20 | 0.999 | 8w | 0.0004 | 49.22 |
| | Reimba | 2 | 0.999 | 5w | 0.0004 | 81.95 |
| | Reimba | 5 | 0.999 | 10w | 0.0004 | 74.24 |
| | Reimba | 10 | 0.999 | 10w | 0.0004 | 55.99 |
| | Reimba | 20 | 0.999 | 10w | 0.0004 | 49.47 |
| WK-m | Rein*for*mer | 2 | 0.99 | 2w | 0.0001 | 79.84 |
| | Rein*for*mer | 5 | 0.99 | 1.5w | 0.0001 | 79.91 |
| | Rein*for*mer | 10 | 0.99 | 1.5w | 0.0001 | 79.82 |
| | Rein*for*mer | 20 | 0.99 | 2w | 0.0001 | 72.25 |
| | Reinconver | 2 | 0.99 | 7w | 0.0001 | 72.28 |
| | Reinconver | 5 | 0.99 | 7w | 0.0001 | 74.09 |
| | Reinconver | 10 | 0.99 | 7w | 0.0001 | 75.88 |
| | Reinconver | 20 | 0.99 | 7w | 0.0001 | 75.38 |
| | Reimba | 2 | 0.999 | 1w | 0.0001 | 77.81 |
| | Reimba | 5 | 0.999 | 1.5w | 0.0001 | 80.03 |
| | Reimba | 10 | 0.999 | 1w | 0.0001 | 75.59 |
| | Reimba | 20 | 0.999 | 1w | 0.0001 | 73.35 |

| env name | model | $K$ | tau | train step | learning rate | normalized score |
|---|---|---|---|---|---|---|
| KC-p | Rein*for*mer | 2 | 0.9 | 20w | 0.0001 | 68.05 |
| | Rein*for*mer | 5 | 0.9 | 20w | 0.0001 | 73 |
| | Rein*for*mer | 10 | 0.9 | 20w | 0.0001 | 74.05 |
| | Rein*for*mer | 20 | 0.9 | 10w | 0.0001 | 66.2 |
| | Reinconver | 2 | 0.99 | 20w | 0.0001 | 65.2 |
| | Reinconver | 5 | 0.99 | 20w | 0.0001 | 34.55 |
| | Reinconver | 10 | 0.99 | 20w | 0.0001 | 65.85 |
| | Reinconver | 20 | 0.99 | 20w | 0.0001 | 65.25 |
| | Reimba | 2 | 0.99 | 6w | 0.0001 | 40.75 |
| | Reimba | 5 | 0.99 | 5w | 0.0001 | 45.1 |
| | Reimba | 10 | 0.99 | 4w | 0.0001 | 29.7 |
| | Reimba | 20 | 0.99 | 2w | 0.0001 | 29.05 |
| MZ-l | Rein*for*mer | 2 | 0.999 | nan | 0.0004 | nan |
| | Rein*for*mer | 5 | 0.999 | 10w | 0.0004 | 64.95 |
| | Rein*for*mer | 10 | 0.999 | 10w | 0.0004 | 62 |
| | Rein*for*mer | 20 | 0.999 | 10w | 0.0004 | 64.99 |
| | Reinconver | 2 | 0.999 | 10w | 0.0004 | 32.69 |
| | Reinconver | 5 | 0.999 | 10w | 0.0004 | 22.45 |
| | Reinconver | 10 | 0.999 | 10w | 0.0004 | 34.74 |
| | Reinconver | 20 | 0.999 | 10w | 0.0004 | 32.39 |
| | Reimba | 2 | 0.999 | 10w | 0.0004 | 59.00 |
| | Reimba | 5 | 0.999 | 5w | 0.0004 | 41.04 |
| | Reimba | 10 | 0.999 | 5w | 0.0004 | 43.59 |
| | Reimba | 20 | 0.999 | 5w | 0.0004 | 43.14 |
| P-h | Rein*for*mer | 2 | 0.9 | 4w | 0.0001 | 62.77 |
| | Rein*for*mer | 5 | 0.9 | 4w | 0.0001 | 75.15 |
| | Rein*for*mer | 10 | 0.9 | 10w | 0.0001 | 68.25 |
| | Rein*for*mer | 20 | 0.9 | 10w | 0.0001 | 71.94 |
| | Reinconver | 2 | 0.99 | 4w | 0.0001 | 73.64 |
| | Reinconver | 5 | 0.99 | 4w | 0.0001 | 82.23 |
| | Reinconver | 10 | 0.99 | 5w | 0.0001 | 76.27 |
| | Reinconver | 20 | 0.99 | 5w | 0.0001 | 75.29 |
| | Reimba | 2 | 0.99 | 4w | 0.0001 | 84.89 |
| | Reimba | 5 | 0.99 | 4w | 0.0001 | 82.91 |
| | Reimba | 10 | 0.99 | 4w | 0.0001 | 97.31 |
| | Reimba | 20 | 0.99 | 4w | 0.0001 | 91.61 |
| P-c | Rein*for*mer | 2 | 0.9 | 5w | 0.0001 | 64.49 |
| | Rein*for*mer | 5 | 0.9 | 5w | 0.0001 | 86.55 |
| | Rein*for*mer | 10 | 0.9 | 5w | 0.0001 | 75.17 |
| | Rein*for*mer | 20 | 0.9 | 5w | 0.0001 | 74.79 |
| | Reinconver | 2 | 0.99 | 5w | 0.0001 | 68.52 |
| | Reinconver | 5 | 0.99 | 5w | 0.0001 | 71.68 |
| | Reinconver | 10 | 0.99 | 5w | 0.0001 | 62.58 |
| | Reinconver | 20 | 0.99 | 5w | 0.0001 | 83.38 |
| | Reimba | 2 | 0.99 | 5w | 0.0001 | 59.60 |
| | Reimba | 5 | 0.99 | 5w | 0.0001 | 71.28 |
| | Reimba | 10 | 0.99 | 5w | 0.0001 | 71.02 |
| | Reimba | 20 | 0.99 | 5w | 0.0001 | 70.57 |

| env name | model | K | tau | learning rate | normalized score |
|---|---|---|---|---|---|
| AT-mp | Rein*for*mer | 2 | 0.999 | 0.0008 | 5.8 |
| | Rein*for*mer | 5 | 0.999 | 0.0008 | 4.2 |
| | Rein*for*mer | 10 | 0.999 | 0.0008 | 3.8 |
| | Rein*for*mer | 20 | 0.999 | 0.0008 | 1.6 |
| | Reinconver | 2 | 0.999 | 0.0008 | 6.2 |
| | Reinconver | 5 | 0.999 | 0.0008 | 7.8 |
| | Reinconver | 10 | 0.999 | 0.0008 | 4.4 |
| | Reinconver | 20 | 0.999 | 0.0008 | 2 |
| | Reimba | 2 | 0.999 | 0.0008 | 5.2 |
| | Reimba | 5 | 0.999 | 0.0008 | 12.4 |
| | Reimba | 10 | 0.999 | 0.0008 | 13.8 |
| | Reimba | 20 | 0.999 | 0.0008 | 15.6 |
| AT-md | Rein*for*mer | 2 | 0.999 | 0.0008 | 2 |
| | Rein*for*mer | 5 | 0.999 | 0.0008 | 3.4 |
| | Rein*for*mer | 10 | 0.999 | 0.0008 | 5.6 |
| | Rein*for*mer | 20 | 0.999 | 0.0008 | 4.2 |
| | Reinconver | 2 | 0.999 | 0.0008 | 5.4 |
| | Reinconver | 5 | 0.999 | 0.0008 | 4.2 |
| | Reinconver | 10 | 0.999 | 0.0008 | 5.2 |
| | Reinconver | 20 | 0.999 | 0.0008 | 2.6 |
| | Reimba | 2 | 0.999 | 0.0008 | 2.6 |
| | Reimba | 5 | 0.999 | 0.0008 | 5 |
| | Reimba | 10 | 0.999 | 0.0008 | 5 |
| | Reimba | 20 | 0.999 | 0.0008 | 9 |

Table 6: The normalized scores of Reinconver and Reimba with and without positional embeddings. Original Reimba and Reinconver did not include positional embedding. The $\Delta$ represents the change in score when positional embedding is added.

| | | | WK-m | | | HC-me | | |
|---|---|---|---|---|---|---|---|---|
| model | K | no_pos | pos | $\Delta$ | no_pos | pos | $\Delta$ |
| Reinconver | 5 | 74.09 | 75.48 | +1.88% | 92.26 | 91.80 | -0.50% |
| Reimba | 5 | 80.03 | 74.73 | -6.62% | 92.91 | 91.57 | -1.44% |

Table 7: window size for Reinconver

| K | 2 | 5 | 10 | 20 |
|---|---|---|---|---|
| window size | 4 | 6 | 10 | 20 |

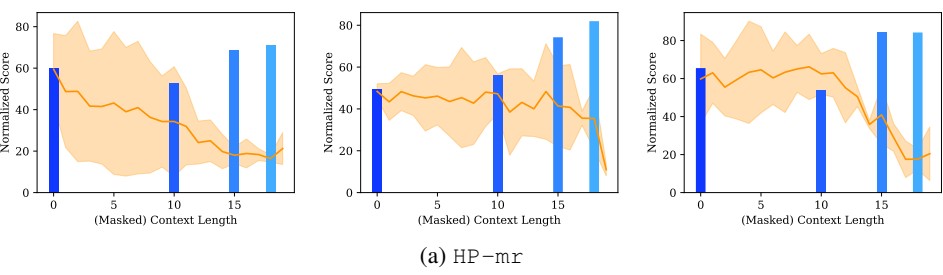

(a) HP-mr

Figure 9: This figure displays the performance of masking the first $(20 - K_1)$ tokens in a sequence model with $K = 20$. We show the averages and corresponding standard deviations of three seeds evaluated in the *HP-mr* environment 10 times (represented by the solid yellow line and its shaded area). Additionally, we compare this with models trained and evaluated normally with a length of $20, 10, 5, 2$ (blue bar values). The horizontal axis increases from left to right as the number of masked tokens increases and the remaining context length $K_1$ decreases.

