# OpenReview forum: "Revisiting the Design Choices in Max-Return Sequence Modeling"
_ICLR.cc/2025/Conference — Submitted to ICLR 2025_

### Official Review · Reviewer_z4Tj · 2024-10-28

**Soundness:** 3
**Presentation:** 2
**Contribution:** 2
**Rating:** 3
**Confidence:** 4

**Summary:**

This paper revisits sequence modeling for offline reinforcement learning (RL) by exploring design choices like architecture and context length across diverse datasets, moving beyond the limited evaluations typically used in this area. The paper makes use of max-return sequence modeling, which predicts maximized returns to guide actions, removing the need for manually set target returns and allowing for broader dataset applicability. Key findings highlight that dataset characteristics often impact performance more than model architecture, providing insights into when sequence modeling or traditional RL methods may be preferable.

**Strengths:**

The max-return sequence modeling framework addresses the limitation of manually specified target returns effectively.

The paper explores nine datasets, three model architectures, and various context lengths, which is a notable improvement over previous work limited to simpler datasets.

The findings on architecture and context length provide useful guidelines for practitioners selecting models for different offline RL tasks.

**Weaknesses:**

To enhance reproducibility, it would be highly beneficial to release code and resources. Specifically, providing implementations of key model components (e.g., max-return sequence modeling framework), data preprocessing scripts, and evaluation protocols would allow researchers to replicate and build on your work effectively. Including guidelines on setting hyperparameters and selecting context lengths for different dataset characteristics could further assist reproducibility and practical application.

Although the paper mentions that dataset characteristics significantly impact model performance, it does not delve deeply into specific dataset factors that might inform architecture or hyperparameter tuning. The analysis of dataset characteristics could be deepened by exploring specific factors that might guide architecture or hyperparameter decisions. For example, aspects such as trajectory length distribution, reward sparsity, and the dimensionality of the state and action spaces could provide insights into selecting context length or architecture. A discussion on how these characteristics might impact model performance would help clarify the tuning process for specific tasks or dataset types.

While the paper makes a valuable effort to explore underrepresented datasets, a more explicit discussion about why specific datasets were chosen and any limitations of these datasets would strengthen the robustness of the study, especially concerning sequence modeling applicability.

Very minor typo at line 060: “Overall, the dataset characteristics of the have greater …”
(you should remove “of the”)

Line 067: This is very awkwardly phrased and should be more clearly stated. As is the first remark on context length being minor to performance:

“The impact of context length on performance is relatively minor.”

Seems contradictory to the following two sentences:

 “A shorter context length is more advantageous for trajectory stitching. Moreover, it is surprising to find that models trained on long sequences perform exceptionally well during inference with short sequences, significantly enhancing their trajectory stitching capabilities.”

**Questions:**

What dataset characteristics specifically correlate with optimal context lengths? The paper mentions variations in context length without thoroughly linking these findings to specific dataset properties. It would strengthen the findings to identify specific dataset characteristics that correlate with optimal context lengths. Again, properties such as trajectory length, state/action space dimensionality, and reward sparsity could be analyzed to determine how they influence context length choices. Exploring these aspects could reveal generalizable patterns for determining optimal context lengths in other datasets and applications, potentially providing valuable tuning guidelines.

Do you anticipate that the observed effects of architecture and context length on performance will hold for other reinforcement learning tasks or domains not included in the study? Any insight into the potential for generalizing these findings would strengthen the conclusions. Or do you have any marginal evidence that might suggest this?

Will you provide code, model checkpoints, or more detailed instructions for setting up experiments? This would help others replicate and validate your findings and would enhance the practical impact of your contributions.

---

### Official Review · Reviewer_zr9C · 2024-11-02

**Soundness:** 3
**Presentation:** 4
**Contribution:** 3
**Rating:** 5
**Confidence:** 4

**Summary:**

The paper presents an interesting perspective on offline RL by applying principles of supervised learning with max-return sequence modeling. This approach claims to sidestep the complexities traditionally associated with RL, such as the fitting of optimal value functions and policy gradient computation.

The idea of treating offline RL as a supervised learning task by focusing on sequence modeling could potentially simplify implementations and increase the applicability of RL in diverse settings.

To me, there are several areas that require clarification. I am happy to adjust my rating.

**Strengths:**

1. Interesting perspective: Supervised sequence modeling is applied to offline RL in an interesting way, in which complexity can be reduced.
2. Comprehensive analysis and experiment: The submission includes thorough experiments across datasets.

**Weaknesses:**

A few issues in writing are raised:
1. consistent abbreviations for datasets in Table 1 and the rest of the manuscript
2. The baseline method implicit QL (IQL) is not cited
3. Missing figure number in Line436

**Questions:**

1. Given that one of the potential advantages of this method over traditional RL is reduced complexity, are the computational resources used reduced as well? More details on this would help understand the method's practical benefits.
2. In Section 5.1, the quality of the datasets, which claims to be the reason affecting the performance against IQL, can be further elaborated.
3. Table 4 suggests a linear relationship between the performance and context length $K$. Could you discuss the reason for choosing to model these relationships linearly? And, if I was not mistaken, in this regression analysis, $K$ values are limited to the discrete support of $\{2,5,10,20\}$, which may not accurately reflect the relationship by the interpolation. Presenting these plots visually assessing this relationship could provide clearer evidence of linearity or indicate the need for a different modeling approach, such as polynomial regression or categorically treating each $K$ value. Alternatively, more detailed justification for the chosen method would help clarify the rationale behind this modeling choice.

---

### Official Review · Reviewer_4Ya6 · 2024-11-03

**Soundness:** 3
**Presentation:** 3
**Contribution:** 1
**Rating:** 3
**Confidence:** 4

**Summary:**

The paper revisits design choices in sequence modelling for offline reinforcement learning, focusing on the Decision Transformer and Max-Return paradigm. It systematically analyses the effect of architectural variations and context lengths across nine datasets, including benchmarks beyond the commonly used Gym environments. The findings highlight that dataset characteristics significantly influence performance, and authors make some high level recommendations based on dataset quality and task setting.

**Strengths:**

- The paper performs an extensive evaluation across a diverse set of datasets, highlighting the variability of performance due to data distribution. An important point that helps compare different methods more fairly across a number of relevant settings
- The work explores multiple architectural designs and context lengths, presenting detailed insights into their performance implications
-  The paper highlights how sequence modelling is better suited for settings with high quality data, whereas classical RL methods are better suited for tasks requiring trajectory stitching. Additionally, it introduces novel findings related to context length and positional embeddings
- The paper is well written and the experiments are described clearly and throughly. It is easy to follow and explains past work well

**Weaknesses:**

- Despite its in-depth analysis, the paper primarily focuses on re-evaluating existing methods rather than introducing significant new ideas or algorithms, limiting its novelty
- The insights provided, although useful, seem incremental (e.g. it was already known that RL is better at trajectory stitching compared to sequence modelling)
- Nit: seems like a word is missing or the sentence at line 59 is wrong
- Nit: I would remove the "quite" at line 62
- Nit: I think it would be useful to briefly introduce trajectory stitching before where it is now (line 115) and closer to where it is first mentioned (line 68)
- Nit: 186 The 3 and the 4 are bold but the 9 is not

**Questions:**

- In the future work, you mention it would be interesting to explore how to better integrate classical RL with sequence modelling to harness the strength of both, do you have any concrete ideas here? I feel like including something like it in this current paper could make it much stronger
- You mention that positional embeddings might hinder trajectory stitching. Could you elaborate on this insight? What types of positional embeddings did you use? Do you think different tasks / datasets might benefit from different types of positional embeddings? (e.g. absolute vs cosine etc.)

---

### Official Review · Reviewer_WgaL · 2024-11-04

**Soundness:** 3
**Presentation:** 3
**Contribution:** 2
**Rating:** 5
**Confidence:** 3

**Summary:**

This paper is an investigation of offline RL and max-return sequence models. Three architectures, Reinformer, Reinconver, and Reimba; 9 standard continuous control offline RL datasets; and different context lengths are investigated. The authors find that with a diverse enough evaluation, no method truly comes out on top. They also find that model and hyperparameter order appears to heavily depend on the environment, and the data quality of the replay buffer. This is particularly visible with the effect of training and inference context length. For the latter, the authors expand on previous results and show that well-thought masking can help models at inference perform well with only few steps of context (even if they are trained with many steps).

**Strengths:**

Investigative papers are important, and this paper does a great job of being systematic & wide, and questioning past claims of performance. This is helpful in understanding where the field must progress.

The paper is fairly well written, and although it would benefit from a bit of proofreading, it was easy to understand and follow. Since this is investigational work, the novelty aspect is less salient, but nonetheless the authors perform some original investigations.

**Weaknesses:**

Maybe the most unsatisfying part is that this paper feels like it scratches the surface of many things, without going really into depth in one of the presented aspects. A next-level version of this paper would focus on one thing only and ask much deeper mechanistic questions; why are things the way they are? What are the right ways to measure that? Does this lead to (potential) novel methods?

For example, the authors could focus on the dataset quality. What happens when we go from `random` to `medium` to `expert`? What happens if we take the `expert` dataset and corrupt it? Take subsets of it of varying quality? Introduce distractors? I think it's fairly obvious to any ML researcher that data quality deeply affects ML models; the better question to answer is what specifically in the data has an impact? In RL we know there's tons of domain-specific structure (e.g. time), so we have the occasion to ask way more specific mechanistic questions of our models.

On one hand, I appreciate this paper and its "retrospectiveness", and it raises important questions. On the other hand, I think a much stronger version of this paper could exist e.g. as I suggest above, and I don't see this paper being a very strong contribution in the sense of being impactful to offline RL researchers other than as a regular call to order for papers to take diverse benchmarking seriously.

**Questions:**

- In 5.2.1, the authors use a "0" token, but presumably the model has never (or rarely?) seen this. Why not just compute the Jacobian or some similar quantity to see the dependence of the prediction on past inputs? Or am I wrong an the models actually do see mask tokens?
- "Thus, the Reinformer pays more attention to global information", this is an interesting observation, but it doesn't translate to supporting a claim the architecture impacts performance. If different architectures "weigh" temporal information differently but have the exact same policy logits, are they really different? There needs to be a more causal/mechanistic claim here.
- There's a fair amount of speculation (especially about environment properties) that feels hidden in the text, for example, the authors claim that "the Reinformer’s focus on global historical trajectory information is particularly adept at considering and utilizing waypoint-related information effectively." This is speculation from (a) Figure 2 (b) and the authors' understanding of `maze-large`. A better, and less speculative, experiment would directly intervene on the agent to test a hypothesis; for example in this case, the agents could be made to "cheat" by being given waypoint information, and if this intervention removes the Reinformer's superiority, then we've learned that the other models are probably worse at modeling this specifically.
- Is the x axis inverted on Figures 3 and 4 and 6b? Table 2 (and the text) suggests e.g. that Reinformer is superior to Reimba on `Maze2d-large`.
- "Correspondingly, a smaller context length aligns more closely with the Markov Decision Process (MDP) framework, and thus performs better." I'm not sure this conclusion naturally follows. If I understand correctly, the argument is that offline RL on a worse & diverse replay buffer means more effective capacity is spent modeling unimportant states, which means that we should show less of them at a time to be closer to a 1-Markov setting. To me this is not obviously true because even with a lower K, the model still sees all the states as part of its training data, and is still spending effective capacity on them regardless of the sequence in which they appear. To me a more plausible reason is that _credit assignment_ is harder if the model has to consider many possible explanations (i.e. K times more) for the output; or in other words with longer inputs the variance is larger. But this could be demonstrated empirically rather than through speculation.

---

### Meta-Review · Area_Chair_sBUW · 2024-12-20

**Metareview:**

This paper revisits the design choices in the sequence modeling approach in offline RL. It carries out systematic investigation in a wide range of benchmarks and compare various architectures. It provides insights on the dependence of performance on context length and architectural choices. However, most reviewers find this work lack of depths in its investigation. The authors did not respond to reviews.

**Additional Comments On Reviewer Discussion:**

No rebuttal or discussion.

---

### Decision · Program_Chairs · 2025-01-22

Reject